# CHARACTERIZING INTRINSIC COMPOSITIONALITY IN TRANSFORMERS WITH TREE PROJECTIONS

**Shikhar Murty**[†]  **Pratyusha Sharma**[‡]  **Jacob Andreas**[‡]  **Christopher D. Manning**[†]
[†]Computer Science Department, Stanford University   [‡]MIT CSAIL
{smurty, manning}@cs.stanford.edu, {pratyusha, jda}@mit.com

## ABSTRACT

When trained on language data, do transformers learn some arbitrary computation that utilizes the full capacity of the architecture or do they learn a simpler, tree-like computation, hypothesized to underlie compositional meaning systems like human languages? There is an apparent tension between compositional accounts of human language understanding, which are based on a restricted bottom-up computational process, and the enormous success of neural models like transformers, which can route information arbitrarily between different parts of their input. One possibility is that these models, while extremely flexible in principle, in practice learn to interpret language hierarchically, ultimately building sentence representations close to those predictable by a bottom-up, tree-structured model. To evaluate this possibility, we describe an unsupervised and parameter-free method to *functionally project* the behavior of any transformer into the space of tree-structured networks. Given an input sentence, we produce a binary tree that approximates the transformer's representation-building process and a score that captures how "tree-like" the transformer's behavior is on the input. While calculation of this score does not require training any additional models, it provably upper-bounds the fit between a transformer and any tree-structured approximation. Using this method, we show that transformers for three different tasks become more tree-like over the course of training, in some cases unsupervisedly recovering the same trees as supervised parsers. These trees, in turn, are predictive of model behavior, with more tree-like models generalizing better on tests of compositional generalization.

## 1 INTRODUCTION

Consider the sentence *Jack has more apples than Saturn has rings*, which you have almost certainly never encountered before. Such *compositionally novel* sentences consist of known words in unknown contexts, and can be reliably interpreted by humans. One leading hypothesis suggests that humans process language according to hierarchical tree-structured computation and that such a restricted computation is, in part, responsible for compositional generalization. Meanwhile, popular neural network models of language processing such as the transformer can in principle, learn an arbitrarily expressive computation over sentences, with the ability to route information between any two pieces of the sentence. In practice, when trained on language data, do transformers instead constrain their computation to look equivalent to a tree-structured bottom-up computation?

While generalization tests on benchmarks (Lake & Baroni, 2018; Bahdanau et al., 2019; Hupkes et al., 2019; Kim & Linzen, 2020, among others) assess if a transformer's *behavior* is aligned with tree-like models, they do not measure if the transformer's *computation* is tree-structured, largely because model behavior on benchmarks could entirely be due to orthogonal properties of the dataset (Patel et al., 2022). Thus, to understand if transformers implement tree-structured computations, the approach we take is based on *directly approximating them* with a separate, tree-structured computation. Prior methods based on this approach (Andreas, 2019; McCoy et al., 2019) require putatively gold syntax trees, which not only requires committing to a specific theory of syntax, but crucially, may not exist in some domains due to syntactic indeterminacy. Consequently, these methods will fail to recognize a model as tree-like if it is tree-structured according to a different notion of syntax. Moreover, all of these approaches involve an expensive training procedure for explicitly fitting a tree-structured model (Socher et al., 2013; Smolensky, 1990) to the neural network.

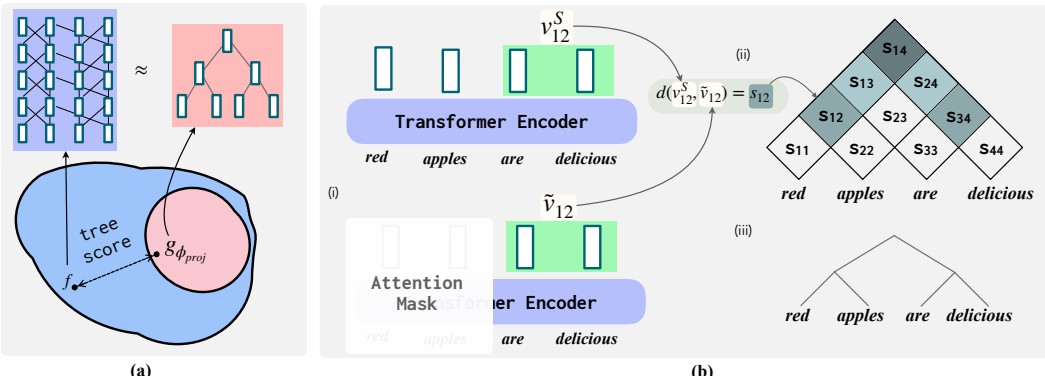

Figure 1: (a) Given a transformer model $f$, our method finds the *tree projection* of $f$ i.e., binary trees corresponding to the tree-structured neural network $g_{\phi_{\text{proj}}}$ (in the space of all tree-structured models) that best approximates the outputs of $f$ on a given set of strings. (b) (i) Given a string, we compute context-free representations ($\tilde{v}_{ij}$) for all spans of the string via attention masking (Section 3). (ii) We use the distance between (average-pooled) context-free and contextual representations ($v_{ij}$) to populate a chart data structure. (iii) We decode a tree structure from chart entries.

Instead, we present a method that is completely unsupervised (no gold syntax needed) and parameter-free (no neural network fitting needed). At a high level, our proposed method *function-ally projects*[1] transformers into the space of all tree-structured models, via an implicit search over the joint space of tree structures and parameters of corresponding tree-structured models (Figure 1). The main intuition behind our approach is to appeal to the notion of *representational invariance*: bottom-up tree-structured computations over sentences build intermediate representations that are invariant to outside context, and so we can approximate transformers with a tree-structured computation by searching for a "bracketing" of the sentence where transformer representations of inter-mediate brackets are maximally invariant to their context. Concretely, the main workhorse of our approach is a subroutine that computes distances between contextual and context-free representations of all spans of a sentence. We use these distances to induce a *tree projection* of the transformer using classical chart parsing (Section 3), along with a score that estimates tree-structuredness.

First, we prove that our approach can find the *best* tree-structured account of a transformer's computation under mild assumptions (Theorem 1). Empirically, we find transformer encoders of varying depths become more tree-like as they train on three sequence transduction datasets, with correspond-ing tree projections gradually aligning with gold syntax on two of three datasets (Section 5). Then, we use tree projections as a tool to predict behaviors associated with compositionality: induced trees reliably reflect contextual dependence structure implemented by encoders (Section 6.1) and both tree scores as well as parsing F1 of tree projections better correlate with compositional generalization to configurations unseen in training than in-domain accuracy on two of three datasets (Section 6.2).

## 2 BACKGROUND

How can we compute the meaning of *red apples are delicious*? Substantial evidence (Crain & Nakayama, 1987; Pallier et al., 2011; Hale et al., 2018) supports the hypothesis that semantic in-terpretation of sentences by humans involves a *tree-structured*, hierarchical computation, where smaller constituents (*red*, *apples*) recursively combine into larger constituents (*red apples*), until we reach the full sentence. Concretely, suppose we have a sentence $S \triangleq \{w_1, w_2, \ldots, w_{|S|}\}$. Let $T$ be a function that returns a binary tree for any sentence $S$, defined recursively as $T(S) \triangleq \langle T(S_{1,j}), T(S_{j+1,|S|}) \rangle$ where $T(S_{a,b})$ refers to a subtree over the span $S_{a,b} \triangleq \{w_a, w_{a+1}, \ldots, w_b\}$. We say that a span $S_{a,b} \in T(S)$ if the node $T(S_{a,b})$ exists as a subtree in $T(S)$. For notational con-venience, we sometimes use $S_l$ and $S_r$ as the left and right subtrees for $T(S)$ i.e., $T(S) = \langle S_l, S_r \rangle$.

---

[1]We provide a *functional* account of the transformer's computation and not a *topological* account, i.e., we are agnostic to whether the attention patterns of the transformer themselves look tree structured—see Appendix C for examples.

**Compositionality in Meaning Representations.** While theories of compositional meaning formation might differ on specifics of syntax, at a high-level, they propose that computing the meaning of $S$ must involve a bottom-up procedure along some syntax tree $T(S)$ of the sentence $S$. Formally, we say that a meaning representation system $m$ is compositional if the meaning $m(s)$ of some expression $s$ is a *homomorphic image* of the syntax of $s$ i.e., $m(s) = \phi(m(s_l), m(s_r))$ for some $\phi$ following Montague (1970). Crucially, we note that such a $\phi$ exists only if $m(s)$ can be fully determined by the contents of $s$, that is, if $m(s)$ is *contextually invariant*. While there are several phenomena that necessarily require a non-compositional context-sensitive interpretation (indexicals, idioms, pronouns, lexical ambiguity among others), compositional interpretation remains a central component in explanations of the human ability to systematically interpret novel sentences.

**Compositionality in Neural Models.** A class of neural networks that are obviously compositional are tree-structured models such as Socher et al. (2013), that obtain *vector representations* of sentences by performing a bottom-up computation over syntax. Specifically, given $S$ and a corresponding binary tree $T(S)$, the output of the tree-structured network $g_\phi$ is defined recursively—for any span $p \in T(S)$, $g_\phi(p, T(p)) \triangleq h_\theta(g_\phi(p_l, T(p_l)), g_\phi(p_r, T(p_r)))$ where $h_\theta : \mathbb{R}^d \times \mathbb{R}^d \mapsto \mathbb{R}^d$ is some feedforward neural network. For leaf nodes $w_i$, $g_\phi(w_i, T(w_i)) \triangleq \eta_{w_i}$, where $\eta_w \in \mathbb{R}^d$ represents the word embedding for $w$. The parameters of the network are $\phi = \{\theta, \eta_{w_1}, \eta_{w_2}, \ldots\}$.

## 3 OUR APPROACH

While tree-structured networks were built to reflect the compositional structure of natural language, they have been superseded by relatively unstructured transformers (Vaswani et al., 2017). How can we measure if the *computation* implemented by a transformer is compositional and tree-like? We start by noting that in any bottom-up tree computation over a sentence, representation of an intermediate constituent depends only on the span it corresponds to, while being fully invariant to outside context. Thus, one way to assess tree-structuredness of a computation over some span is to measure *contextual invariance* of the resulting representation. Consequently, we construct a tree-structured approximation of a transformer's computation over a sentence by searching for a bracketing of the sentence where spans have maximal contextual invariance.

### 3.1 SPAN CONTEXTUAL INVARIANCE

Suppose $f$ is a transformer model that produces contextual vectors of words in $S$ as $f(S) \triangleq \{\boldsymbol{v}_{w_1}^S, \boldsymbol{v}_{w_2}^S, \ldots, \boldsymbol{v}_{w_{|S|}}^S\}$ where $\boldsymbol{v}_w^S$ is a contextual vector representation of $w$. Given a span $p$, let $\boldsymbol{v}_p^S$ be the span representation of the contextual vectors of words in $p$, $\boldsymbol{v}_p^S = \sum_{w \in p} \boldsymbol{v}_w^S$. Similarly, let $\tilde{\boldsymbol{v}}_p$ be a *context-free* representation of the span $p$. For transformers, we obtain context-free representations through a simple attention masking scheme. In particular, to obtain $\tilde{\boldsymbol{v}}_p$, we apply a "T-shaped" attention mask and take the pooled representation of the words in $p$ at the final layer (Figure 2). The mask ensures that attention heads do not attend to tokens outside of $p$ *after an optional threshold layer*[2]

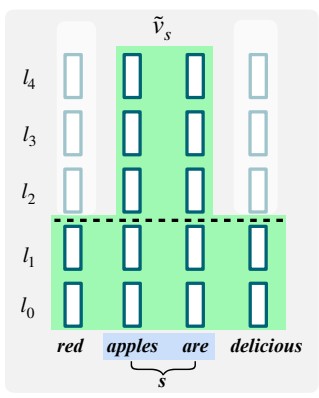

Figure 2: We use a T-shaped attention mask with a threshold layer to obtain approximate context-free vectors for transformers.

We define span contextual invariance (SCI) of a span $p$ in the sentence $S$ as $\text{SCI}(S, p) \triangleq d(\boldsymbol{v}_p^S, \tilde{\boldsymbol{v}}_p)$ for some distance function $d$. Similarly, we define the cumulative SCI score for a tree $T$ to be:

$$\text{SCI}(S, T) \triangleq \sum_{s \in T} d(\boldsymbol{v}_p^S, \tilde{\boldsymbol{v}}_p). \qquad (1)$$

---

[2]This procedure outputs vectors that are entirely context-free only if the threshold is exactly 0, but we find that tuning the threshold layer often leads to significantly better induced parses.

## 3.2 Computing Tree Projections by minimizing SCI

Consider the collection of strings, $\mathcal{D} = \{(S)\}$, and some function $T$ that produces binary trees for any $S \in \mathcal{D}$. The cumulative error from approximating outputs of the transformer $f$ with outputs of a tree-structured network $g_\phi$ structured according to $T$ can be written as

$$\mathcal{L}(f, g_\phi, T) \triangleq \sum_{S \in \mathcal{D}} \sum_{p \in T(S)} d(g_\phi(p, T(p)), \boldsymbol{v}_p^S). \tag{2}$$

Suppose we are interested in finding the best tree-structured approximation to $f$ over all possible trees i.e. a configuration of tree structures and corresponding model parameters that best approximate the transformer's behavior. We define this as the *exact tree projection* of $f$,

$$\phi_{\text{proj}}, T_{\text{proj}} \triangleq \arg\min_{\phi, T} \mathcal{L}(f, g_\phi, T). \tag{3}$$

**Theorem 1.** $\min_{\phi, T} \mathcal{L}(f, g_\phi, T) \leq \sum_{S \in \mathcal{D}} \min_{T(S)} \text{SCI}(S, T(S))$. *In other words, the best tree structured approximation to $f$ has an error upper bounded by cumulative* SCI *scores.*

In general, finding tree projections involves a joint search over all discrete tree structures $T(S)$ as well as over continuous parameters $\phi$, which is intractable. However, we substantially simplify this search using Theorem 1, since the upper bound depends only on parses $T(S)$ and properties of the transformer, and can be exactly minimized for a given $f$ in polynomial time, with efficient parsing algorithms. We minimize this upper bound itself to approximately recover the best tree-structured approximation to $f$, over all choices of trees and parameters. The output of this minimization is an approximate tree projection,

$$\widehat{T}_{\text{proj}}(S) = \arg\min_{T(S)} \text{SCI}(S, T(S)) \tag{4}$$

for every $S \in \mathcal{D}$. Under a mild assumption[3], SCI minimization leads to tree projections *exactly*.

**Assumption 1.** *Let $S_p$ denote the collection of sentences that contain the span $p$. Then, for every span $p$, we have $\min_{\boldsymbol{v}} \sum_{S \in S_p} d(\boldsymbol{v}_p^S, \boldsymbol{v}) = \sum_{S \in S_p} d(\boldsymbol{v}_p^S, \tilde{\boldsymbol{v}}_p)$. That is, context-free vectors minimize the cumulative distance to their contextual counterparts.*

**Corollary 1.1.** *Under Assumption 1, $\min_{\phi, T} \mathcal{L}(f, g_\phi, T) = \sum_{S \in \mathcal{D}} \min_{T(S)} \text{SCI}(S, T(S))$. Moreover, $T_{proj}(S) = \arg\min_{T(S)} \text{SCI}(S, T(S))$ for any $S \in \mathcal{D}$.*

### 3.3 Measuring Intrinsic Compositionality

SCI minimization provides two natural ways to measure intrinsic compositionality of $f$ on $\mathcal{D}$. To measure tree-structuredness, we use

$$t_{\text{score}} \triangleq \frac{\sum_{S \in \mathcal{D}} \mathbb{E}_T \text{SCI}(S, T) - \text{SCI}(S, \widehat{T}_{\text{proj}}(S))}{|\mathcal{D}|}, \tag{5}$$

which computes the averaged SCI score of induced trees, normalized against the expected SCI score under a uniform distribution over trees. We find normalization to be necessary to prevent our method from spuriously assigning high tree-structuredness to entirely context-free encoders (that have high SCI scores for *all* trees). When gold syntax $T_g$ is available, we use

$$t_{\text{parseval}} \triangleq \text{PARSEVAL}(\widehat{T}_{\text{proj}}, T_g, \mathcal{D}), \tag{6}$$

to measure bracketing F1 score (PARSEVAL; Black et al. (1991)) score of $\widehat{T}_{\text{proj}}$ against $T_g$ on $\mathcal{D}$.

## 4 Experimental Setup

Our experiments[4] are organized as follows. First, we show that on 3 sequence transduction tasks, transformers of varying depths become more tree-like over the course of training, and sometimes learn tree projections that progressively evolve towards ground truth syntax. Then, we show how tree projections can be used to assess various model behaviors related to compositionality.

---

[3]Figure 9 in the Appendix shows that this assumption approximately holds in practice.

[4]Code and data will be available here

**Datasets.** We consider three datasets (Table 1) commonly used for benchmarking compositional generalization—COGS (Kim & Linzen, 2020), M-PCFGSET (Hupkes et al., 2019) and GeoQuery (Zelle & Mooney, 1996). COGS consists of automatically generated sentences from a context-free grammar paired with logical forms, split into in-domain examples (for training) and a compositionally challenging evaluation set. M-PCFGSET is a slightly modified version [5] of PCFGSET (Hupkes et al., 2019), where inputs are a nested sequence of expresssions that specify a unary or binary operation over lists. The objective is to execute the function specified by the input to obtain the final list. We focus on the "systematicity split" for measuring compositional generalization. Finally, GeoQuery consists of natural language queries about US geography paired with logical forms. To measure compositional generalization, we use the "query" split from Finegan-Dollak et al. (2018).

**Implementation Details.** We use greedy top down chart parsing to approximately minimize SCI. In particular, we use SCI scores for all $O(|S|^2)$ spans of a string $S$ to populate a chart data structure, which is used to induce a tree by minimizing SCI via a top down greedy procedure (see Algorithm 1 in Appendix), similar to Stern et al. (2017). Our procedure outputs a tree and simultaneously returns normalized SCI score of the tree, computing a sampling estimate of expected SCI score (Equation 5).We train transformer encoder-decoder models with encoders of depths $\{2, 4, 6\}$ and a fixed decoder of depth 2. We omit 6-layer transformer results for GeoQuery as this model rapidly overfit and failed to generalize, perhaps due to the small size of the dataset. We choose a shallow decoder to ensure that most of the sentence processing is performed on the encoder side. We train for 100k iterations on COGS, 300k iterations on M-PCFGSET and 50k iterations on GeoQuery. We collect checkpoints every 1000, 2000 and 500 gradient updates and use the encoder at these checkpoints to obtain parses as well as tree scores. In all experiments, $d$ is cosine distance i.e., $d(\boldsymbol{x}, \boldsymbol{y}) = 1 - \frac{\boldsymbol{x}^\top \boldsymbol{y}}{\|\boldsymbol{x}\|\|\boldsymbol{y}\|}$. All transformer layers have 8 attention heads and a hidden dimensionality of 512. We use a learning rate of 1e-4 (linearly warming up from 0 to 1e-4 over 5k steps) with the AdamW optimizer. All accuracies refer to exact match accuracy against the gold target sequence. For all seq2seq transformers, we tune the threshold layer based on $t_{\text{parseval}}$.

| | Inputs | Outputs |
|---|---|---|
| i. | *The ball was found* 
 *A cookie was blessed* | `ball(x₁) AND find.theme(x₃, x₁)` 
 `cookie(x₁) AND bless.theme(x₃, x₁)` |
| ii. | *copy interleave_second reverse shift H13 C19 H9 O20* 
 *repeat interleave_second interleave_first S1 E3 W3 N11 H4 Y3* | `H9 H13 O20 C19` 
 `L8 E1 R13 T12 E1 T12 L8 E1 R13 T12 E1 T12` |
| iii. | *Which state has the lowest population density?* 
 *What is the population density of Wyoming?* | `(A, _smallest(B, (_state(A), _density(A, B))))` 
 `(A, (_density(B, A), _const(B, _stateid(wyoming))))` |

Table 1: Example $(x, y)$ pairs from COGS (i), M-PCFGSET (ii) and GeoQuery (iii). See Appendix B for more details on pre-processing as well as dataset statistics.

## 5 TRAINED TRANSFORMERS IMPLEMENT A TREE-LIKE COMPUTATION

How does intrinsic compositionality of a transformer encoder evolve during the course of training on sequence transduction tasks? To study this, we plot $t_{\text{score}}$ (*how tree-like is a model?*) and $t_{\text{parseval}}$ (*how accurate is the tree projection of a model?*) of encoder checkpoints throughout training. As a comparison, we track how well a supervised probe recovers syntax from encoders—that is, we train a 1 layer transformer decoder to autoregressively predict linearized *gold* parse trees of $S$ from transformer outputs $f(S)$ at various points of training, and measure the PARSEVAL score of probe outputs ($p_{\text{parseval}}$) on a test set.

**Results.** We plot $t_{\text{parseval}}$ and $t_{\text{score}}$ over the course of training in Figure 3. *We observe that 7/8 encoders gradually become more tree-like* i.e., increase $t_{\text{score}}$ over the course of training, with the 4 layer transformer on GeoQuery being the exception. Interestingly, we note that $t_{\text{parseval}}$ also increases over time for all encoders on COGS and M-PCFGSET suggesting that the *tree projection of trained transformers progressively becomes more like ground-truth syntax*. In other words, all encoders trained on COGS and M-PCFGSET learn a computation that is gradually more "syntax aware". Can supervised probing also reveal this gradual syntactic enrichment? We plot PARSEVAL score of

---

[5] see Appendix B for details.

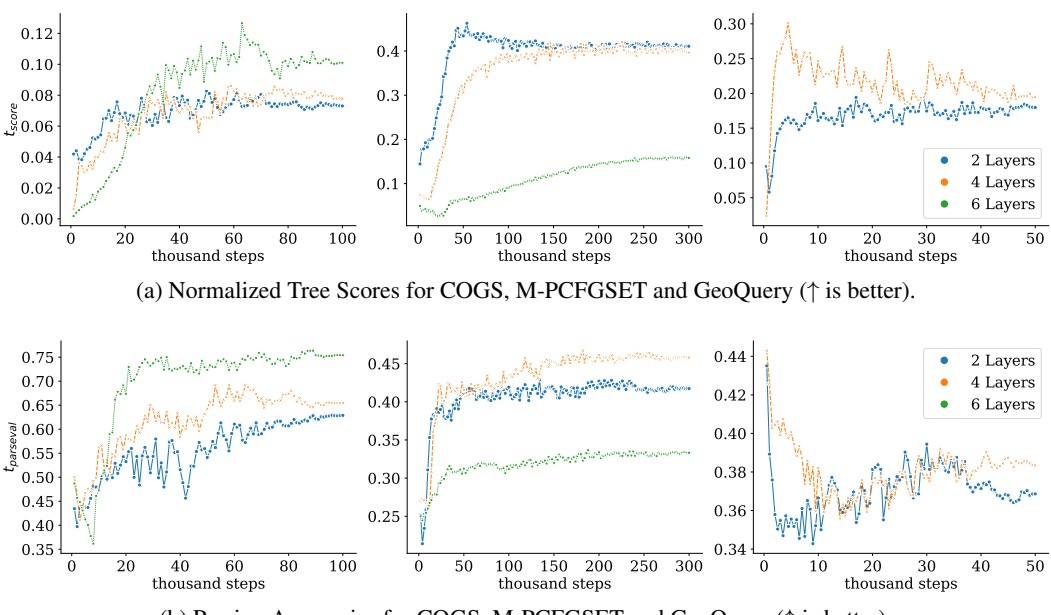

(a) Normalized Tree Scores for COGS, M-PCFGSET and GeoQuery ($\uparrow$ is better).

(b) Parsing Accuracies for COGS, M-PCFGSET and GeoQuery ($\uparrow$ is better).

Figure 3: We plot $t_{\text{score}}$ and $t_{\text{parseval}}$ by computing approximate tree projections at various checkpoints. 7/8 models become more tree-structured (increased $t_{\text{score}}$) and all models on COGS and M-PCFGSET learn tree projections that gradually align with ground truth syntax (increased $t_{\text{parseval}}$).

parse trees predicted by the probe on held out sentences ($p_{\text{parseval}}$) in Figure 4—while $p_{\text{parseval}}$ does improve over time on both COGS and M-PCFGSET, we observe that all checkpoints after some threshold have similar probing accuracies. We quantitatively compare gradual syntactic enrichment by computing the spearman correlation between $t_{\text{parseval}}$ ($p_{\text{parseval}}$) and training step and find that $\rho_{p_{\text{parseval}}}$ is significantly smaller than $\rho_{t_{\text{parseval}}}$ for both datasets. Interestingly, we also find that our unsupervised procedure is able to produce better trees than the *supervised* probe on M-PCFGSET as observed by comparing $p_{\text{parseval}}$ and $t_{\text{parseval}}$. Overall, we conclude that supervised probing is unable to discover latent tree structures as effectively as our method.

**How does supervisory signal affect compositionality?** Could a purely self-supervised objective (i.e., no output logical form supervision) also lead to similar emergent tree-like behavior? To test this, we experiment with training the transformer encoder with a masked language modeling objective, similar to Devlin et al. (2019) for COGS and GeoQuery. Concretely, for every $S$, we mask out 15% of input tokens and jointly train a transformer encoder and a 1 layer feedforward network, to produce contextual embeddings from which the feedforward network can decode word identities for masked out words. As before, we collect checkpoints during training and plot both $t_{\text{parseval}}$ and $t_{\text{score}}$ over time in Figure 5. We find that $t_{\text{parseval}}$ does not improve over time for any of the models. Additionally, we find that $t_{\text{score}}$ increases for all models on GeoQuery, but only for the 2 layer model on COGS. Taken together, these results suggest that under the low data regime studied here, transformers trained with a self-supervised objective do not learn tree-structured computations.

## 6  TREE PROJECTIONS AND MODEL BEHAVIOR

Given $S$, and corresponding contextual vectors $f(S)$, the *contextual dependence structure* captures the dependence between contextual vectors and words in $S$ i.e., how much does $\boldsymbol{v}_{w_i}^S$ change when $w_j$ is perturbed to a different word. Contextual dependence structure is important for assessing compositional behavior. For instance, consider the span $p = red\ apples$ appearing in some sentences. If the contextual vectors for $p$ has large dependence on outside context, we expect the model to have poor generalization to the span appearing in *novel contexts* i.e., poor compositional generalization.

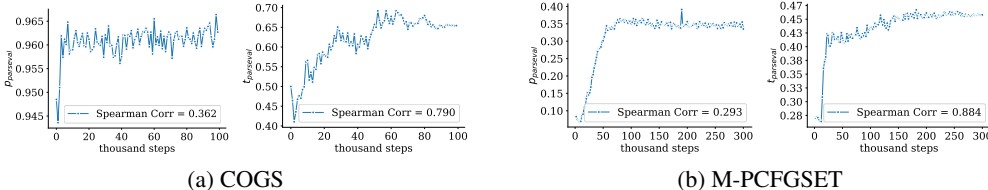

(a) COGS                    (b) M-PCFGSET

Figure 4: We plot $p_{\text{parseval}}$ and $t_{\text{parseval}}$ over time for the 4 layer transformer encoder on COGS and M-PCFGSET. We find that $t_{\text{parseval}}$ improves gradually over time suggesting that the model becomes more "syntax aware". Such gradual syntax enrichment is not uncovered well by the probe since all checkpoints after 4000 (for COGS) and 50000 (for M-PCFGSET) iterations have similar $p_{\text{parseval}}$.

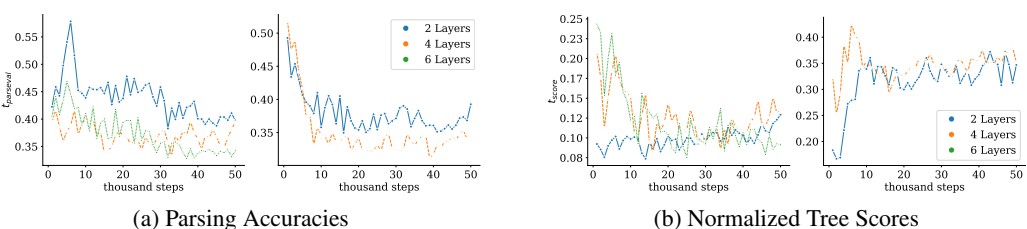

(a) Parsing Accuracies              (b) Normalized Tree Scores

Figure 5: We plot $t_{\text{parseval}}$ and $t_{\text{score}}$ at various checkpoints for models trained with a masked language modeling objective on COGS (first) and GeoQuery (second). Only 2/5 models become tree-structured and none learn tree projections aligned with gold syntax, suggesting that self-supervision may fail to produce tree-like computation in a relatively low data regime.

We first show that tree projections reflect the contextual dependence structure implemented by a transformer. Next, we show that both $t_{\text{score}}$ and $t_{\text{parseval}}$ are better predictors of compositional generalization than in-domain accuracy.

## 6.1 INDUCED TREES CORRESPOND TO CONTEXTUAL DEPENDENCE STRUCTURE

Intuitively, greedily decoding with a SCI populated chart makes split point decisions where resulting spans are maximally invariant with one other. Thus, for a given constituent $c$ and a word $w \in c$, we expect $\boldsymbol{v}_w^S$ to depend more on words within the same constituent than words outside the constituent. Thus, we compare the change in $\boldsymbol{v}_w^S$ when another word inside $c$ is perturbed (*in-constituent* perturbations) to the change when a word outside $c$ is perturbed (*out-of-constituent* perturbations), where word perturbations are performed by adding gaussian noise to corresponding word vectors in layer 0 (see Figure 6). We ensure that both perturbations are made to words at the same *relative distance* from $w$. As a control, we also compute changes to $\boldsymbol{v}_w^S$ when perturbations are made with respect to constituents from random trees.

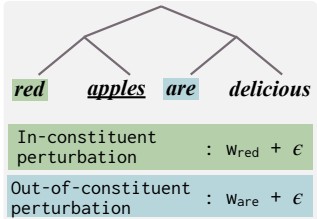

Figure 6: For word $w$ (*apples*) in constituent $c$, an in-constituent perturbation adds noise $\epsilon \sim \mathcal{N}(0, 0.01)$ to another word's vector within $c$ (*red*) while an out-of-constituent perturbation adds noise to a word vector at same relative distance outside $c$ (*are*).

**Setup and Results.** We sample 500 random inputs from each of COGS, M-PCFGSET and GeoQuery and consider encoders from all transformer models. We obtain the *mean $L_2$ distance* between the contextual vector of $w$ in the original and perturbed sentence for in-constituent perturbations ($\Delta_{ic}$) and out-of-constituent perturbations ($\Delta_{oc}$) and plot the relative difference between the two in Figure 7. For 6/8 models, in-constituent perturbations result in larger $L_2$ changes than out-of-constituent perturbations (statistically significant according to a two-sided $t$-test, $p < 10^{-4}$). Meanwhile, when constituents are chosen according to random trees, changes resulting from both perturbations are similar. Overall, this suggests that *induced trees reflect the contextual dependence structure learnt by a transformer*.

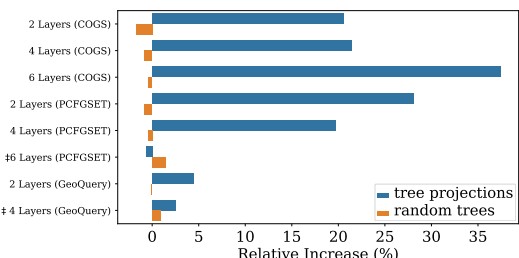

Figure 7: We measure the mean $L_2$ distance in the contextual vector of words when *in-constituent* and *out-of-constituent* words are perturbed. We plot the relative difference between $\Delta_{ic}$ and $\Delta_{oc}$ when constituents are obtained from tree projections (in blue). As a control, we also compute $\Delta_{ic}$ and $\Delta_{oc}$ when constituents are chosen from random trees (in orange). For all models except those marked with ‡, *in-constituent* perturbations lead to significantly (as measured by a *t*-test, $p < 10^{-5}$) larger change to contextual vectors compared to *out-of-constituent* perturbations.

## 6.2 TREE-STRUCTUREDNESS CORRELATES BETTER WITH GENERALIZATION THAN IN-DOMAIN ACCURACY

We study the connection between compositionality and generalization for the 4 layer transformer encoder on COGS and GeoQuery [6]. On each dataset, we train the model with 5 different random seeds and collect checkpoints every 1000/500 iterations. For each checkpoint, we measure accuracy on the in-domain validation set (*IID acc*) and accuracy on the out-of-domain compositional generalization set (*CG acc*). Additionally, we also compute $t_{\text{parseval}}$ and $t_{\text{score}}$ for the encoders at each of these checkpoints. To measure the relationship between compositionality and generalization, we compute the spearman correlation between $t_{\text{parseval}}$ ($t_{\text{score}}$) and *CG acc* and denote that as $\rho^{\text{CG}}_{t_{\text{parseval}}}$ ($\rho^{\text{CG}}_{t_{\text{score}}}$). As a comparison, we also compute the correlation between *IID acc* and *CG acc* ($\rho^{\text{CG}}_{\text{IID}}$).

**Results.** We plot the relationship between various properties and generalization along with corresponding correlations in Figure 8. In general, we expect both *IID acc* and *CG acc* to improve together over time, and so it is unsurprising to see that $\rho^{\text{CG}}_{\text{IID}} > 0$. Moreover, for COGS, both $t_{\text{parseval}}$ and $t_{\text{score}}$ increase over time, and so it is expected that both $\rho^{\text{CG}}_{t_{\text{parseval}}}$ and $\rho^{\text{CG}}_{t_{\text{score}}}$ are positive. Crucially, however, we find that both $\rho^{\text{CG}}_{t_{\text{parseval}}}$ and $\rho^{\text{CG}}_{t_{\text{score}}}$ are greater than $\rho^{\text{CG}}_{\text{IID}}$ on both COGS and GeoQuery. Thus, tree-like behavior ($t_{\text{score}}$) as well as the *right* tree-like behavior ($t_{\text{parseval}}$) are better predictors of compositional generalization than in-domain accuracy. This result gives simple *model selection* criteria to maximize CG accuracy in the absence of a compostional generalization test set (true for most practical scenarios)—given a collection of checkpoints with similar in-domain accuracies, choose the checkpoint with highest $t_{\text{score}}$ or $t_{\text{parseval}}$ (if syntactic annotations are available) to get the model with best generalization behavior, in expectation.

## 7 RELATED WORK

**Measuring Linguistic Structure.** A common analysis tool for assessing a model's competence in a specific linguistic phenomenon is *behavioral testing* (Linzen et al., 2016; Marvin & Linzen, 2018; Ribeiro et al., 2020), where the model's performance on a curated test set is used as the measure of competence. Widely used in prior work to assess compositionality of neural models (Lake & Baroni, 2018; Bahdanau et al., 2019; Yu & Ettinger, 2020), behavioral tests are inherently *extrinsic*, since they are agnostic to whether the model implements an appropriately constrained, tree-like computation. While most prior approaches for assessing intrinsic compositionality (Andreas, 2019; McCoy et al., 2019) require putatively gold syntax trees, our proposed approach does not require any pre-determined ground truth syntax, since we search over the space of *all* possible trees to find the best tree structure that approximates a transformer's computation.

**Tree-structured Neural Networks.** Inspired by the widely accepted belief that natural language is mostly tree-structured (Chomsky, 1957), there have been several attempts to construct tree shaped

---

[6]IID acc perfectly predicts generalization for M-PCFGSET so we omit it in these experiments

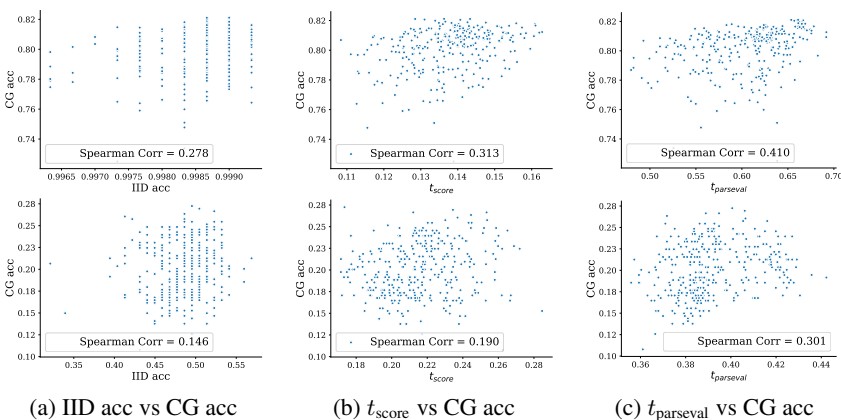

(a) IID acc vs CG acc      (b) $t_{\text{score}}$ vs CG acc      (c) $t_{\text{parseval}}$ vs CG acc

Figure 8: We plot the spearman correlation between (a) *IID acc* and *CG acc*, (b) $t_{\text{score}}$ and *CG acc*, (c) $t_{\text{parseval}}$ and *CG acc*. We find that both $t_{\text{parseval}}$ and $t_{\text{score}}$ correlate better with generalization than in-domain accuracy. All correlations are statistically significant ($p$-values $< 10^{-3}$)
.

neural networks for various NLP tasks, such as Recursive Neural Networks (Socher et al., 2013), Tree RNNs (Tai et al., 2015), Recurrent Neural Network Grammars (Dyer et al., 2016), Neural Module Networks (Andreas et al., 2016), Ordered Neuron (Shen et al., 2019) among others. These approaches have largely been superseded by transformers (Vaswani et al., 2017), often pre-trained on a large corpus of text (Devlin et al. (2019), *inter alia*). We show that transformers, though not explicitly tree-structured, may still learn to become tree-like when trained on language data.

**Invariances and Generalization.** The general problem of studying model performance under domain shifts has been widely studied under domain generalization (Blanchard et al., 2011). When domain shift is a result of changing feature covariates only, an effective strategy for domain generalization is to learn *domain invariant representations* (Muandet et al., 2013; Ganin et al., 2016). We apply the notion of domain invariance in the context of compositional generalization, and posit that models that produce span representations that are more contextually invariant can generalize better to inputs where the span appears in a novel context, which is precisely the motivation behind SCI.

## 8 CONCLUSION

When trained on language data, how can we know whether a transformer learns a compositional, tree structured computation hypothesized to underlie human language processing? While extrinsic behavioral tests only assess if the model is capable of the same generalization capabilities as those expected from tree-structured models, this work proposes an *intrinsic* approach that directly estimates how well a parametric tree-structured computation approximates the model's computation. Our method is unsupervised and parameter-free and provably upper bounds the representation building process of a transformer with any tree-structured neural network, effectively providing a *functional projection* of the transformer into the space of all tree structured models. The central conceptual notion in our method is *span contextual invariance* (SCI) that measures how much the contextual representation of a span depends on the context of the span vs. the content of the span. SCI scores of all spans are plugged into a standard top-down greedy parsing algorithm to induce a binary tree along with a corresponding tree score. From experiments, we show that tree projections uncover interesting training dynamics that a supervised probe is unable to discover—we find that on 3 sequence transduction tasks, transformer encoders tend to become *more tree-like* over the course of training, with tree projections that become *progressively closer to true syntactic derivations* on 2/3 datasets. We also find that tree-structuredness as well as parsing F1 of tree projections is a better predictor of generalization to a compositionally challenging test set than in-domain accuracy i.e., given a collection of models with similar in-domain accuracies, select the model that is most tree-like for best compositional generalization. Overall, our results suggest that making further progress on human-like compositional generalization might require inductive biases that encourage the emergence of latent tree-like structure.

## 9 ACKNOWLEDGEMENTS

SM was funded by a gift from Apple Inc. JA is supported by the MIT Quest for Intelligence through a grant from Liberty Mutual Insurance. CM is a fellow in the CIFAR Learning in Machines and Brains program. We thank Ekin Akyürek, Marco Tulio Ribeiro, John Hewitt, Alexis Ross and members of the Stanford NLP group for feedback on early drafts on the paper.

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

# A  PROOFS

**Lemma 1.** $\mathcal{L}(f, g_{\phi^*}, T) \leq \sum_{S \in \mathcal{D}} \text{SCI}(S, T(S))$

*Proof.* Let $l(f, g_\phi, S, T) \triangleq \sum_{s \in T(S)} d(g_\phi(s, T(s)), \boldsymbol{v}_s^S)$ for any $S \in \mathcal{D}$, where $g$ is a tree-structured network indexed by $\phi \in \mathbb{R}^p$. The overall error of $g_\phi$ on $\mathcal{D}$ is

$$\mathcal{L}(f, g_\phi, T) = \sum_{S \in \mathcal{D}} l(f, g_\phi, S, T). \tag{7}$$

Let $\phi^* \triangleq \arg\min_\phi \mathcal{L}(f, g_\phi, T)$. Next, consider $\hat{\phi} \in \mathbb{R}^p$ such that $g_{\hat{\phi}}(s, T(s)) = \tilde{\boldsymbol{v}}_s$ for all $s \in \mathcal{D}$. Such a $\hat{\phi}$ always exists for large enough $p$, since there exists a unique $\tilde{\boldsymbol{v}}_s$ for any $p$ given $\mathcal{D}$ and $f$. Clearly, $l(f, g_{\hat{\phi}}, S, T) = \sum_{s \in T(S)} d(\boldsymbol{v}_s^S, \tilde{\boldsymbol{v}}_s)$. By definition, we have

$$\mathcal{L}(f, g_{\phi^*}, T) \leq \mathcal{L}(f, g_{\hat{\phi}}, T) \tag{8}$$

$$= \sum_{S \in \mathcal{D}} \sum_{s \in T(S)} d(\boldsymbol{v}_s^S, \tilde{\boldsymbol{v}}_s) = \sum_{S \in \mathcal{D}} \text{SCI}(S, T(S)). \tag{9}$$

$\square$

**Theorem 1.** $\min_{\phi, T} \mathcal{L}(f, g_\phi, T) \leq \sum_{S \in \mathcal{D}} \min_{T(S)} \text{SCI}(S, T(S))$. *In other words, the best tree structured approximation to $f$ has an error upper bounded by cumulative* SCI *scores.*

*Proof.* We have

$$\min_{\phi, T} \mathcal{L}(f, g_\phi, T) = \min_T \min_\phi \mathcal{L}(f, g_\phi, T) \tag{10}$$

For any given $T$, we have $\min_\phi \mathcal{L}(f, g_\phi, T) \leq \sum_{S \in \mathcal{D}} \text{SCI}(S, T(S))$. Thus minimizing both sides with respect to $T$, we have

$$\min_T \min_\phi \mathcal{L}(f, g_\phi, T) \leq \min_T \sum_{S \in \mathcal{D}} \text{SCI}(S, T(S)) \tag{11}$$

$$= \sum_{S \in \mathcal{D}} \min_{T(S)} \text{SCI}(S, T(S)) \tag{12}$$

$\square$

Under Assumption 1 and Theorem 1, we have the proof for Corollary 1.1 which we present next.

**Corollary 1.1.** *Under Assumption 1,* $\min_{\phi,T} \mathcal{L}(f, g_\phi, T) = \sum_{S \in \mathcal{D}} \min_{T(S)} \text{SCI}(S, T(S))$. *Moreover,* $T_{proj}(S) = \arg\min_{T(S)} \text{SCI}(S, T(S))$ *for any* $S \in \mathcal{D}$.

*Proof.* Let $s_T$ be the collection of all spans that occur as a constituent for some $T(S)$ where $S \in \mathcal{D}$. We have

$$\mathcal{L}(f, g_\phi, T) = \sum_{S \in \mathcal{D}} \sum_{s \in T(S)} d(g_\phi(s, T(s)), \boldsymbol{v}_s^S) \tag{13}$$

$$= \sum_{s \in s_T} \sum_{S \in S_s} d(g_\phi(s, T(s)), \boldsymbol{v}_s^S). \tag{14}$$

Now, using Assumption 1, we note that

$$\sum_{S \in S_s} d(g_\phi(s, T(s)), \boldsymbol{v}_s^S) \geq \min_{\boldsymbol{v}} \sum_{S \in S_s} d(\boldsymbol{v}, \boldsymbol{v}_s^S) = \sum_{S \in S_s} d(\tilde{\boldsymbol{v}}_s, \boldsymbol{v}_s^S). \tag{15}$$

Combining Equation 15 and Lemma 1, we have

$$\min_\phi \mathcal{L}(f, g_\phi, T) = \sum_{S \in \mathcal{D}} \text{SCI}(S, T(S)) \tag{16}$$

Now, we have

$$T_{\text{proj}} = \arg\min_T \left[ \min_\phi \mathcal{L}(f, g_\phi, T) \right] = \arg\min_T \sum_{S \in \mathcal{D}} \text{SCI}(S, T(S)) \tag{17}$$

Thus, $T_{\text{proj}}(S) = \arg\min_{T(S)} \text{SCI}(S, T(S))$ □

Next, we consider specific examples of distance metric $d$, and what Assumption 1 implies for context-free vectors $\tilde{\boldsymbol{v}}_s$.

**Example A.1.** *Suppose $d$ is the euclidean $L_2$ distance i.e., $d(\boldsymbol{x}, \boldsymbol{y}) = \|\boldsymbol{x} - \boldsymbol{y}\|$. Then, Assumption 1 requires that $\tilde{\boldsymbol{v}}_s = \frac{1}{|Ss|} \sum_{S \in S_s} \boldsymbol{v}_s^S$*

*Proof Sketch.* We have $\boldsymbol{v}_s^* = \arg\min_v \sum_{S \in S_s} d(\boldsymbol{v}_s^S, \boldsymbol{v}) = \arg\min_v \sum_{S \in S_s} \|\boldsymbol{v} - \boldsymbol{v}_s^S\|$. Setting derivatives with respect to $\boldsymbol{v}$ to 0, we have $\boldsymbol{v}_s^* = \frac{1}{|S_s|} \sum_{S \in S_s} \boldsymbol{v}_s^S$ □

**Example A.2.** *Let $d$ be the cosine distance of $\boldsymbol{x}$ and $\boldsymbol{y}$ i.e., $d(\boldsymbol{x}, \boldsymbol{y}) = 1 - \frac{\boldsymbol{x}^\top \boldsymbol{y}}{\|\boldsymbol{x}\|\|\boldsymbol{y}\|}$. Then, Assumption 1 requires that $\tilde{\boldsymbol{v}}_s = \frac{1}{|S_s|} \sum_{S \in S_s} \frac{\boldsymbol{v}_s^S}{\|\boldsymbol{v}_s^S\|}$*

*Proof Sketch.* We have $\boldsymbol{v}_s^* = \arg\min_v \sum_{S \in S_s} d(\boldsymbol{v}_s^S, \boldsymbol{v}) = \arg\max_v \sum_{S \in S_s} \frac{\boldsymbol{v}^\top \boldsymbol{v}_s^S}{\|\boldsymbol{v}\|\|\boldsymbol{v}_s^S\|} = \arg\max_v \frac{\boldsymbol{v}^\top}{\|\boldsymbol{v}\|} \left( \sum_{S \in S_s} \frac{\boldsymbol{v}_s^S}{\|\boldsymbol{v}_s^S\|} \right)$. Thus, $\boldsymbol{v}_s^* = k \sum_{S \in S_s} \frac{\boldsymbol{v}_s^S}{\|\boldsymbol{v}_s^S\|}$ for any $k > 0$ □

## B  DATASET PREPROCESSING

Dataset statistics are in Table 2.

**COGS.**  We use the standard train, validation and test splits provided by Kim & Linzen (2020), where we use the "gen" split as our test set. The validation set is drawn from the same distribution as the training data, while the test set consists of compositionally challenging input sentences.

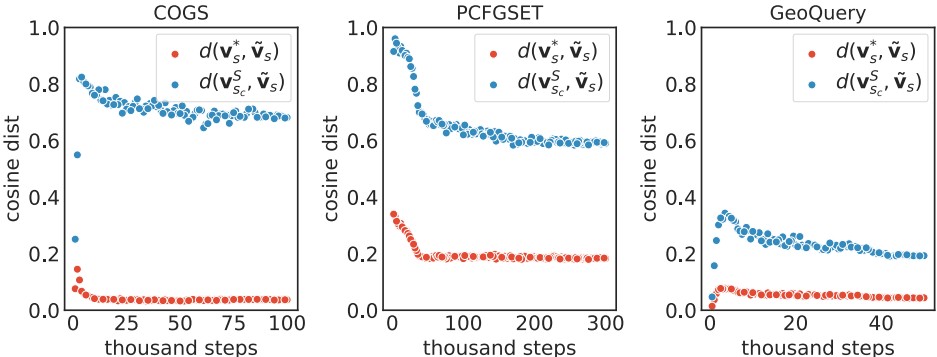

Figure 9: We plot $d(\boldsymbol{v}_s^*, \tilde{\boldsymbol{v}}_s)$ for randomly sampled spans at various points during training. As a control, we also plot $d(\boldsymbol{v}_{s_c}^S, \tilde{\boldsymbol{v}}_s)$ for a random span $s_c$. We observe that for COGS and GeoQuery, the distance between the optimal $\boldsymbol{v}_s^*$ and $\tilde{\boldsymbol{v}}_s$ eventually becomes less than 0.05. We conclude that the conditions of Assumption 1 approximately hold true for 2/3 datasets.

| Dataset | Train | Validation | Test |
|---|---|---|---|
| COGS | 24115 | 3000 | 21000 |
| M-PCFGSET | 65734 | 16434 | 10175 |
| GeoQuery | 434 | 109 | 334 |

Table 2: Dataset Statistics

**M-PCFGSET.** We make two modifications to the PCFGSET dataset. First, we remove commas from expressions so that the model is forced to implictly learn to correctly partition the input expression for a correct intrepretation. To ensure that a unique parse exists even without commas, we additionally ensure that all lists have exactly 2 elements. For instance, the expression `append A B C, D E F` is modified into `append A B E F` that has the unique interpretation `append([A, B], [E, F])` since all lists have exactly 2 elements. Second, we replace the `remove_first` and `remove_second` operations with `interleave_first` and `interleave_second`, where the `interleave` operation takes two lists (say *A B* and *C D*) and interleaves them to either produce *A C B D* or *C A D B*. This modification ensures that intermediate constituents in the expression are not discarded, similar to how intermediate constituents are almost never discarded in natural language utterances.

**GeoQuery.** We use the pre-processed JSON files corresponding to the query split from (Finegan-Dollak et al., 2018). We create an 80/20 split of the original training data, to create an IID validation set.

## C   FUNCTIONAL VS. TOPOLOGICAL TREE-STRUCTUREDNESS

We emphasize that our approach finds a functional tree approximation to a transformer, and not a topological one. That is, we fit a separate, tree structured neural network to vector representations from a transformer, instead of decoding a tree-structure from the attention patterns. As a result, our definition of tree-structuredness does not restrict the transformer's attention pattern to be necessarily tree structured (see Figure 10 for examples).

## D   ANALYZING INDUCED TREE STRUCTURES

We choose the checkpoint with best bracketing F1 score on the training split for all our datasets, and compute corresponding bracketing F1 scores on the IID validation set in Table 3. As a baseline, we compare with standard constituency parsing baselines: LBranch (choosing a completely left branching tree), RBranch (choosing a completely right branching tree) and Random (choosing a

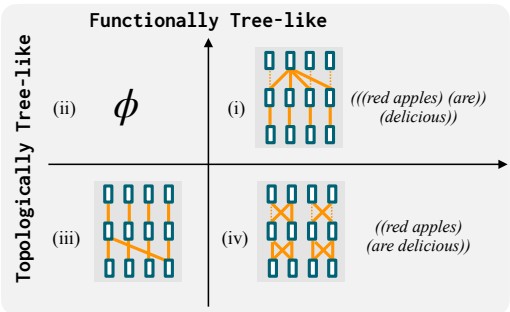

Figure 10: We show 3 instances of computations implemented by a transformer on the input *red apples are delicious* along with tree projections our method outputs for each instance. We divide the space of possibilities into 4 quadrants. In quadrant-(i), we show an instance that is both topologically as well as functionally tree-like. quadrant-(ii) is empty, since no transformer can be topologically tree-like but not a good functional approximation to a tree. In quadrant-(iii) we show a transformer that is either topologically nor functionally tree-like. Finally, in quadrant-(iv), we show a transformer that is functionally tree-like but does not resemble a tree structure topologically.

| Method | COGS | M-PCFGSET | GeoQuery |
|--------|------|-----------|----------|
| TreeProjections | **75.6** | 46.5 | 48.0 |
| Random | 44.9 | 26.8 | 41.0 |
| LBranch | 30.2 | 17.9 | 26.0 |
| RBranch | 75.2 | **58.1** | **69.7** |

Table 3: Parsing accuracies

random binary tree). Interestingly, we find that the trees discovered by our approach on COGS beats RBranch, which is a competitive constituency parsing baseline for English.

---

**Algorithm 1** Tree Projections via greedy SCI minimization

1: **function** TREEPROJECTION($S, f$)
2:     **return** TREEPROJECTIONRECURSE($S, f, 1, |S|$)
3: **end function**

4: **function** TREEPROJECTIONRECURSE($S, f, i, j$)
5:     **if** $i = j$ **then**
6:         ▷ *leaf node*
7:         **return** $w_i, 0$;
8:     **else**
9:         ▷ *greedily select split point to minimize* SCI *of resulting constituents*
10:         $k^* \leftarrow \arg\min_{k \in [i,j)}[\text{SCI}(S_{i,k}) + \text{SCI}(S_{k+1,j})]$;
11:         $s_{k^*} \leftarrow \text{SCI}(S_{i,k^*}) + \text{SCI}(S_{k^*+1,j})$;
12:         ▷ *select a random split point for normalization*
13:         $s_b \leftarrow \text{SCI}(S_{i,k_b}) + \text{SCI}(S_{k_b+1,j}), k_b \sim U[i, j-1]$;
14:         ▷ *Recursively call the function to get a tree structure and score for left span*
15:         $S_l, ts_l \leftarrow$ TREEPROJECTIONRECURSE($S, f, i, k^*$);
16:         ▷ *Recursively call the function to get a tree structure and score for the right span*
17:         $S_r, ts_r \leftarrow$ TREEPROJECTIONRECURSE($S, f, k^* + 1, j$);
18:         **return** $\langle S_l, S_r \rangle, s_b - s_{k^*} + ts_l + ts_r$
19:     **end if**
20: **end function**

---

