# OpenReview forum: "Characterizing intrinsic compositionality in transformers with Tree Projections"
_ICLR.cc/2023/Conference — ICLR 2023 poster_

### Official Review · Reviewer_4grv · 2022-10-25

**Confidence:** 4
**Correctness:** 3
**Technical Novelty And Significance:** 2
**Empirical Novelty And Significance:** 2
**Recommendation:** 6

**Clarity, Quality, Novelty And Reproducibility:**

The conclusion found by the paper is not new/surprising to me. Some previous work has already discussed that neural models (including pre-trained models but not limited to transformers) in natural language processing have the ability to learn linguistic structures, e.g., Shen et al. (2019), Coenen et al. (2019), Hewitt and Manning (2019). The important point is how we can utilize this to help the downstream tasks, and I cannot see so much discussed in this paper.

Reference
[1] Ordered Neurons: Integrating Tree Structures into Recurrent Neural Networks. Yikang Shen, Shawn Tan, Alessandro Sordoni and Aaron Courville. ICLR 2019.
[2] Visualizing and Measuring the Geometry of BERT. Andy Coenen, Emily Reif, Ann Yuan, Been Kim, Adam Pearce, Fernanda Viégas and Martin Wattenberg. Neurips 2019.
[3] A Structural Probe for Finding Syntax in Word Representations. John Hewitt and Christopher D. Manning. NACCL 2019.

**Strength And Weaknesses:**

Strength
- The tree projections require no extra training and parameters and can be applied to any transformer-based models to tree-structured models.
- The experiments on compositional generalization reveal some of the non-obvious facts for transformer models' behavior in predicting compositional structures.

Weaknesses
- The conclusion found by the tree projections, i.e., transformer learns a compositional, tree-structured computation is actually not a new finding, which can be found in many previous works.
- The tree projection here is more like a tool for ablation studies. I can not tell how we can use this tree projection to improve downstream tasks such as compositional generalization.

**Summary Of The Paper:**

This paper proposes an unsupervised and parameter-free method that functionally projects transformer models into the space of all tree-structured models. By using this method, they show that transformers for three different tasks become more tree-like over the course of training, in some cases unsupervisedly recovering the same trees as supervised parsers. Finally, they use tree projections as a tool to predict behaviors associated with compositionally, where they find that induced trees reliably reflect contextual dependence structure implemented by encoders and both tree scores as well as parsing F1 of tree projections better correlate with compositional generalization to configurations unseen in training than in-domain accuracy.

**Summary Of The Review:**

This paper provides an interesting tree projection to reveal the transformer's ability to learn tree structure during training. However, the conclusion is not very surprising given that some previous work has already discussed similar findings. I agree that compositionally is an important linguistic property in natural language processing, but NLP people care more about how we can utilize compositionally in downstream tasks instead of whether the model can learn compositionally during training. I recommend the author incorporate this tree projection into enhancing the model's generalizability to compositional generalization, e.g., under domain shift or limited training data.

---

> ### Author Response · Authors · 2022-11-13
> **Thank you for your review!**
>
> We clarify your concerns below:
>
> * **“Transformer learns compositional tree-structured computation is not a new finding”**: As Reviewer mpue points out, this is the first tool that measures *functional tree-structuredness* of transformers (i.e. how well can a tree-structured network approximate a transformer, see Eq. 3). On the other hand, all of the papers cited by 4grv are based on probing, which is about whether information about syntax exists within the weights of a network such that it can be recovered by a probe, which may be possible *even if the network’s computation is not tree structured*. In-fact, experiments in our paper directly show this: From Figure-4(a) a probe can be trained to find gold syntax in a model even if a model is not structured according to it (comparing the left and right plots of Figure-4a). And on the flipside, even if a model is tree-structured, a probe may not be able to find syntax simply by examining model embeddings. Next, we make some more observations about our work vs probing:
>     - As you mention, the problem of tree projections is completely unsupervised with respect to syntax , and is simply tasked with finding some binary tree (may or may not be ground truth syntax) structured network that approximates the behavior of the transformer, while probing works necessarily look for ground truth syntax in model parameters.
>     - More importantly, no probing works makes the connection between compositional generalization (an important problem of great practical interest) and internal tree-structuredness.
> * **How can we utilize this to help the downstream tasks?**: We direct 4grv to experiments in Section-6 which look at the relationship between tree-structuredness and behavior. In particular, from experiments in Section 6.2, we find that tree-structuredness can be used as a property to perform model selection for optimal compositional generalization: given a collection of models with same IID performance, simply choosing the most tree-like model leads to best compositional generalization (based on comparing correlations). This has great practical utility, since previous papers on compositional generalization (Csordas et al. 2021) have commented that early stopping on IID validation set is sub-optimal for generalization since performance continues to go up. For a more quantitative claim, we ask 4grv to kindly look at the general response where we show that early stopping based on our proposed tool can improve compositional generalization by as much as 11 accuracy points for a 6 layer transformer encoder.

---

### Official Review · Reviewer_7etS · 2022-10-25

**Confidence:** 4
**Correctness:** 2
**Technical Novelty And Significance:** 2
**Empirical Novelty And Significance:** 2
**Recommendation:** 3

**Clarity, Quality, Novelty And Reproducibility:**

The paper is clearly written. Methods to assess how the amount of syntactic structure in neural models have been proposed in the past, however there are some differences from previous work: this work does not assume any notion of syntax and looks for the braketing that best approximates context-invariance.

**Strength And Weaknesses:**

The main weakness of the paper is its motivation. Specifically:

* The paper starts by explaining that hierarchical structure is necessary for humans to understand novel utterances, and discusses the concept of composionality. It goes on to introduce context-invariance as a crucial ingredient of it: the meaning of an expression does not depend on anything outside the syntactic tree that the expression spans. While the paper mentions notable exceptions to this assumption: multiple word meanings, idioms, etc, it still contrasts compositional interpretation with context-invariance (first paragraph of page 3). I believe that is wrong. It is clear that both play an important role: there is a certain amount of context-invariance required for efficient generalization but language is far from being context-invariant. The two are not at odds with each-other and there is no reason to believe that humans and computational models can’t do both effectively.

* Following this, the paper sets out to measure the context-invariance in the behaviour of transformers. However it’s not clear what questions does the paper aim to answer. Do we want transformers to be context-invariant? No. Do we want transformers to show compositional generalization? Yes, but that should be measured as part of a real-world task, not an artificial task. That is because real world language tasks require both compositional generalization and context-dependence. For example machine translation is a task where models fail to capture context dependence (wrong word meaning, too-literal translations), and it can be argued they show too much compositional generalization.

* These issues continue in the experimental section: the transformer models investigated are trained on small data sets, in some cases using synthetically-generated language and structure-biased tasks such as predicting logical forms. As such the results do not say much about language in general or about real-world language applications.



**Summary Of The Paper:**

The paper proposes a method to project the “behavior” of a transformer model into a binary tree. Crucial to this is the concept of contextual invariance: the representation of a tree node does not depend on anything outside its children. Specifically the method approximates a bracketing of the sentence such that nodes in this tree are maximally invariant under this definition. Given contextual and context-free representations of a span within a transformer model, the distance between them is taken to measure the context-invariance of that span.

The paper goes on to show that tree-like structure is stronger as model train, and that in two of the three data sets used, the trees obtained match a gold syntax. The degree of the tree "match" also correlates with compositional properties of the models as measured on test sets for compositional generalization.


**Summary Of The Review:**

The paper does not address a clear research question: it proposes a method to find a sentence braketing using transformer representations, such that is maximizes the context-invariance of the representations (the representation of a span depends only on its words and not on outside context). It analyzes this braketing and the degree to which it approximates the transformer representations across three tasks used for compositional generalization. Compositional generalization should not be studied in isolation but in the context of real language tasks that require both generalization and context-awareness. Even if the paper had done this, I do not see the impact of method introduced in the paper or of the results. I also find the terminology used throughout the paper miss-leading (such as “tree-like computation” or “tree-structuredness”), as it contrasts hierarchical structure with context-dependence when it’s clear that both are necessary ingredients for modeling language.

---

> ### Author Response · Authors · 2022-11-13
> **Thank you for your review**
>
> Thank you for your thoughtful review. First, we’d like to clearly restate the motivation and objectives of this work. The objective is not to take a stance about whether we want models to be tree-like or not. Our objective is simply to understand *if models tend to become more tree-like, when trained on language data*. Our experiments simply show the result that transformer models do become functionally tree-like when trained on language data. In particular, we find that being more tree-structured also leads to better generalization. In other words, regardless of the status of hierarchical structure in natural language, we show that some property $P$ leads to better generalization to a large class of OOD generalizations This is an interesting finding independent of the fact that $P$ is, in this case, tree-structuredness.
>
> Second, we clarify that GeoQuery is indeed a dataset of natural language text, and is not synthetic: it consists of real questions collected from users about US geography and has been extensively used for semantic parsing.
>
> Do we want transformers to be fully context-invariant? The tools and methods from this work can gracefully handle the trade-off between being tree-like (context-invariant) and context-dependent. To see why, we would like to point 7etS to the general response where we clarify that our approach does not penalize models for context-sensitive processing in the first few layers. Finally, we would like to point 7etS to the definition of the normalized SCI score which we use for assessing tree-structuredness where we note that for a completely context-free transformer, the normalized SCI score will be exactly 0. In summary, we argue that many of the concerns raised by 7etS are already accounted for via the design of the T-shaped attention mask used for computing context free vectors.

---

### Official Review · Reviewer_2Wg3 · 2022-10-26

**Confidence:** 2
**Correctness:** 4
**Technical Novelty And Significance:** 3
**Empirical Novelty And Significance:** 3
**Recommendation:** 6

**Clarity, Quality, Novelty And Reproducibility:**

I found the paper clear and well written, the proposed method seems novel and is explained well enough to reproduce.


**Strength And Weaknesses:**

+ The paper is well written and was relatively easy for me to understand.
+ Theorem 1 is helpful in formalizing how the proposed framework projects transformers to the space of tree structured computations
+ The results appear convincing, with some minor caveats, mentioned below

- I wish the authors would spend more time motivating why this is an interesting problem, and if there are other ways to explain generalization of transformers. Ultimately, the tree computation hypothesis is just one possible explanation for generalization, I'm curious how to know how this work fits into the broader landscape of work
- I found section 6.2 interesting, can the authors expand on this point? Unless I'm misunderstanding, this is a central motivation for the paper and requires further explanation.


**Summary Of The Paper:**

The authors hypothesize that transformers generalize to unseen sentences by implicitly constructing a tree-structured object bottom-up from inputs. They attempt to understand whether the transformer does indeed perform tree-structured computations by approximating them with with a tree. They introduce a novel method to construct tree representations using a procedure that computes distances between contextual and context-free representations of all subsequences in a sentence.


**Summary Of The Review:**

The paper is well written and clear, I think the ideas presented are novel, specifically how to project transformers to the space of trees. I would recommend accepting.

---

> ### Author Response · Authors · 2022-11-13
> **Thanks for your review!**
>
> We thank Reviewer 2Wg3 for their review! We answer your questions below:
>
> **Why study this problem?**
> * We're interested in understanding what inductive biases and internal computational structures neural sequence models implement. As discussed in the response to Reviewer 4grv, probing is inadequate for this purpose because it doesn’t make functional approximations (see Eq.3 and our general response).
> * There is lots of evidence that a bias toward tree-structured computation is important for modeling human sentence processing and structured data like programs, but no existing tools for characterizing tree-structured-ness of model computations and its relation with generalization; we provide the first tool for measuring functional tree-structuredness and the first characterization of its relation to generalization.
>
> **More section 6 details:** At a high level, we propose a certain property $P$ of models, and show that this property has a stronger correlation with generalization than IID accuracy. This means that property $P$ can be used to select the model that has best compositional generalization  among models that are equally good on IID data. In particular, $P$ refers to how tree-like the model is based on closeness to the space of tree-structured neural networks, and how accurate is the tree-projection to some ground truth syntax (Fig-1).  To our knowledge, no other property of neural networks has been shown in prior works to correlate well with compositional generalization. In the general response, we show that we can improve compositional generalization by as much as 11 accuracy points via this model selection strategy.

---

### Official Review · Reviewer_mpue · 2022-10-27

**Confidence:** 3
**Correctness:** 3
**Technical Novelty And Significance:** 4
**Empirical Novelty And Significance:** 4
**Recommendation:** 8

**Clarity, Quality, Novelty And Reproducibility:**

* Clarity: I enjoyed reading the methodology described in the paper. But the experiments and analyses (sec 5 and 6) are quite difficult to follow. Partly because, as mentioned above, I think section 6.1 should be before section 5.

* Quality: The paper should have a significant impact to the community working on analysing compositionality of DL models. However, some arguments (mentioned above) raised in the paper should be carefully examined.

* Originality: The idea and method proposed in the paper is novel and very interesting.

**Strength And Weaknesses:**

The paper tackles the challenging problem with a novel and very interesting idea. The paper makes choices reasonably. First of all, instead of topological tree-structuredness, the paper is looking for *functional* one. This is a novel view for how to understand the compositionality behaviour of a model, and easily turns the problem to optimization. Next, the concept of *span contextual invariance* does reflect the principle of compositionality, and is thus well linguistically motivated. I found the implementation for estimating the tree projection is smart and technically sound. Especially, it can employ the power of the parsing method by Stern et al 2017.

I however found T-shape masking and threshold layer quite tricky, and can't find detail implementation in the paper. It is unclear, e.g. in fig 2, where a threshold layer is put when the number of encoder layers is varying from 2 to 6. And in footnote 2, what does it mean by "tuning the threshold layer"?

The flow of the paper can be more logical. Section 5 uses the proposed t_score and t_parseval to measure tree-structuredness. The key question here is: how good can they measure compositionality? Up to that point, the paper doesn't seem to answer that. And thus it's difficult to see how the conclusions raised in section 5 are reliable. The reader has to wait until section 6.1 for "... induced trees reflect the contextual dependence structure learnt by a transformer."

However, it still doesn't answer why the proposed t should be more reliable than p_parseval. Right below Fig 3 "we conclude that supervised probing is unable to discover latent tree structures as effectively as our method." Why is it bad to converge too fast here? The paper even doesn't show the *true* tree-structuredness improving rate.




**Summary Of The Paper:**

The paper proposes a method to measure how much compositional-ness a transformer model behaviours. The whole work relies on the hypothesis, that inner composition should be independent from outer context. Relying on that hypothesis, the paper proposes the concept of *span contextual invariance* and how to measure it. That leads to a method searching for a tree that is *closet* to the computation of a transformer performing on an utterance. Then, two ways are introduced to measure tree-structuredness, one with and one without a gold tree.

Theoretically, the paper proves that under a mild condition,  minimizing the span contextual invariance yields exact tree projection. Empirically, the paper demonstrates how the two metrics can help to unveil the compositional-ness of transformers. The paper also presents some analyses, especially one on how tree-structureness is correlated to generalisation.

**Summary Of The Review:**

I enjoyed reading the paper and recommend to accept it. The paper has high quality, with novel and very interesting ideas. The work is solid, experiments and analyses are quite thoughtful.

The paper can improved by
* adding more details about implementation (T-shape and threshold layer)
* re-flow some experiments / arguments

---

> ### Author Response · Authors · 2022-11-13
> **Thank you for your comments!**
>
> Thank you for your positive comments on our work! We make clarifications about your questions below:
>
> **Details about the T-shaped attention mask**: We give details for how the threshold is selected in the general response.
>
> **Paper flow + clarity**: Thanks for the feedback! We will restructure the paper to better motivate these metrics by moving Experiment 6.1 closer to the definitions.
>
> **Why is $t_{score}$ better than $t_{parseval}$?** We note that the main difference between the two metrics is that $t_{score}$ measures how tree-like a transformer’s computation is, since it is explicitly computing distance between the transformer’s computation and the best tree-structured network (Fig-1(a)). On the other hand, $t_{parseval}$ measures how well the tree projection approximates ground truth syntax. We believe that $t_{score}$ is better suited for a domain where there is no one notion of ground truth syntax, since it does not require a ground truth parse tree. This can be especially useful when there is syntactic indeterminacy e.g. dealing with PP attachment ambiguity such as “She issued a challenge to voters” or coordination ambiguities (“delicious cookies and milk”).
>
> **Why is it bad for the supervised probe to converge fast?**  In plots of Figure-4, we note that the performance of a supervised probe ($p_{parseval}$) very quickly converges within the first 4000 iterations for COGS. On the other hand, $t_{parseval}$ (which measures whether the computation is actually organized according to gold syntax) continues to improve. This means that a probe can find ground truth syntax in models even if the computations of the model are not organized according to the ground truth syntax (this is measured by $t_{parseval}$). Secondly, since $p_{parseval}$ quickly converges, it is unable to distinguish between checkpoints with varying levels of tree-structuredness. Thus, to understand if a model is functionally structured according to a tree, a supervised probe is inadequate.

---

### Author Response · Authors · 2022-11-13
**General Response**

We thank all reviewers for their responses. Overall, mpue, 2Wg3 and 4grv found our paper to be interesting (4grv, mpue), novel (2Wg3, mpue) with solid experiments and significant impact to the community (mpue). On the other hand, 7etS raises issues with the motivation for studying latent tree structure in NLP models because of the presence of context-sensitive phenomena in language, 4grv asks how this work is different from “probing”  / how tools from this paper can be used to improve task performance, and mpue asks for some details about the t-shaped attention mask. We clarify these concerns below, and address other concerns in individual responses to reviewers.

**Motivation for this work:** In this work, we ask whether the computation function implemented by a transformer resembles the hierarchical, tree-like structure of natural language. And if so, are models that are more tree-like better at generalizing to compositionally novel inputs than ones that aren’t? We would like to emphasize that we are the first to consider the problem of finding tree projections of a model i.e., *functionally approximating* some neural network with a separate tree-structured model by trying to fit a tree-structured model to the outputs of the neural network as shown in Eq. 3 in our paper. This is a completely different notion from probing which simply attempts to see if the _information_ about syntax is present, and not whether the computations themselves are organized according to trees. Next, we are the first to make the connection between compositional generalization (a well-known failure mode of modern machine learning models) and  tree-structuredness, which also has never been studied before. Of particular interest is the key result where more tree-structured models are better at compositional generalization (see Section-6 and the general response), which immediately gives a method to do model selection / early stopping, especially for compositional generalization datasets where early stopping on an IID validation dataset is found to be sub-optimal.

**How does the T-shaped attention mask work?** While natural language is mostly tree-like, there are notable exceptions that require context-sensitive interpretation (some of which we mention in the paper). **To provide a trade-off between context-sensitive interpretation and a tree-like interpretation (mentioned by 7etS)**, our work uses a “T-shaped” attention mask (Figure-2) for computing context-free vectors. For instance, consider a sentence like “The bank ….” where the meaning of bank depends on context. The T-shaped mask allows the vector of bank to attend to all of the words in the context upto a given threshold layer (to perform necessary context-sensitive processing). Thus, a model that has context-sensitive processing in its first few layers is *not penalized by our approach*, and so *our method can gracefully handle the tradeoffs between context-sensitivity and tree structure*. We note that this design decision was heavily motivated by early work on analysing attention patterns in BERT (Clark et al. 2019) where the authors find that the first few layers have a more diffused attention pattern, possibly to perform context-sensitive processing such as disambiguating word senses.

Finally, we provide some clarifications on how this threshold is chosen / implemented (as requested by mpue).
* Implementation: When computing context free vectors for a span $p$, all attention heads in the transformer encoder till threshold layer $l$ are allowed to attend to all words. After threshold layer $l$, attention heads are constrained to only attend to words within the span $p$.

* How is the threshold chosen? Given a transformer encoder with $L$ layers, we compute SCI (and  $t_{\text{parseval}}$)  by applying the T-shaped mask with threshold layer $l \in \{0, 1, 2, \ldots, L/2\}$.  We then choose the threshold that maximises $t_{\text{parseval}}$ on the training set.

**How can we improve task performance with tree projections?** To put the experiments of Section 6.2 to improve performance, we conducted an experiment on COGS where we early stopped based on an IID validation set, vs. our metric $t_\text{parseval}$. After early stopping, we measure compositional generalization of the chosen trained model, and report results in the following Table, where we find that using our metric to early stop results in consistent improvements in compositional generalization, with over a 11 point improvement on COGS with a 6 layer transformer. We note that our approach provides a principled way to early stop for improved compositional generalization, especially since others in the community (Csordas et al. 2021) note that early stopping on IID validation set results in poor performance.

| Model      | Early stop on IID validation |  Early stop on $t_{parseval}$ |
| ------- | ------| ------|
| 2 layers| 81.23 | 82.03 |
| 4 layers| 80.30 | 81.02 |
| 6 layers| 67.46 | 78.62 |

---

### Decision · Program_Chairs · 2023-01-20

**Decision:**

Accept: poster

**Justification For Why Not Higher Score:**

The paper's audience is likely a subset of the NLP community with limited applicability to the wider ICLR audience. The work also presents an analysis tool and doesn't take the extra step of showing how to improve Transformer models with their finding.

**Justification For Why Not Lower Score:**

The work presents a novel and interesting approach to analyse and understand the computation in Transformer models and to show that is tree-structured without the doubts that exist with probe based approaches.

**Metareview: Summary, Strengths And Weaknesses:**

This paper proposes an unsupervised and parameter-free method that functionally projects Transformer models into the space of all tree-structured models. Crucial to this, the authors assume the hypothesis that inner composition should be independent from outer context. Relying on this, the paper proposes the concept of span contextual invariance and ways to measure it. This results in a method that searches for a tree that is closest to the computation of a Transformer and then two ways are introduced to measure tree structuredness.

Strengths:
* The paper is well written.
* Theorem 1 helped formalize how the work projects Transformers to the space of tree structured computations.
* The approach is novel and interesting.
* The tree projections require no extra training and can be applied to any Transformer-based models.
* Experiments on compositional generalization reveal some insights into Transformer models' behaviour.

Weaknesses:
* The flow of the paper could be more logical.
* The problem could be more well motivated. The tree structured computation hypothesis is only one possible explanation for generalization. What is special about this avenue?
* It's somewhat already assumed that Transformers learn a compositional tree-structured computation.
* The tree projection approach here is mostly an analysis tool, it would be interesting if this could be directly used to improve downstream task performance.
* The approach presents context-invariance as an important ingredient for compositionality even though there are many exceptions to this in language.

**Note From Pc:**

if the above contains the word "oral" or "spotlight" please see: "oral" presentation means -> notable-top-5% and "spotlight" means -> notable-top-25%. As stated in our emails, we are disassociating presentation type from AC recommendations

**Summary Of Ac-Reviewer Meeting:**

In the reviewer meeting, the high-level points were that compositionality is interesting but the results in the paper were not very surprising. The results would likely appeal to a subset of the NLP community but may have a smaller audience in the ICLR community. However, the writing and experiments are strong and the tree projection method is novel and interesting. The reviewers recommend the authors find more practical ways to use the tree projection approach to improve results.